# Relation of abnormal cardiac stress testing with outcomes in patients undergoing renal transplantation

**Kelsey Anderson[1], Chirag Bavishi[2], Dhaval Kolte[3], Reginald Gohh[4], James A. Arrighi[2], Philip Stockwell[2], J. Dawn Abbott[2]***

**1** Warren Alpert Medical School, Brown University, Providence, RI, United States of America,
**2** Cardiovascular Institute, The Warren Alpert Medical School at Brown University, Providence, RI, United States of America, **3** Division of Cardiology, Massachusetts General Hospital, Harvard Medical School, Boston, MA, United States of America, **4** Renal Division, Dept of Internal Medicine, Warren Alpert Medical School at Brown University, Providence, RI, United States of America

* Jabbott@lifespan.org

## Abstract

Cardiovascular risk stratification is often performed in patients considered for renal transplantation. In a single center, we sought to examine the association between abnormal stress testing with imaging and post-renal transplant major adverse cardiovascular events (MACE) using multivariable logistic regression. From January 2006 to May 2016 232 patients underwent renal transplantation and 59 (25%) had an abnormal stress test result. Compared to patients with a normal stress test, patients with an abnormal stress test had a higher prevalence of dyslipidemia, diabetes mellitus, obesity, coronary artery disease (CAD), and heart failure. Among those with an abnormal result, 45 (76%) had mild, 10 (17%) moderate, and 4 (7%) severe ischemia. In our cohort, 9 patients (3.9%) had MACE at 30-days post-transplant, 5 of whom had an abnormal stress test. The long-term MACE rate, at a median of 5 years, was 32%. After adjustment, diabetes (OR 2.37, 95% CI 1.12–5.00, p = 0.02), CAD (OR: 3.05, 95% CI 1.30–7.14, p = 0.01) and atrial fibrillation (OR: 5.86, 95% CI 1.86–18.44, p = 0.002) were independently associated with long-term MACE, but an abnormal stress test was not (OR: 0.83, 95% CI 0.37–1.92, p = 0.68). In conclusion, cardiac stress testing was not an independent predictor of long-term MACE among patients undergoing renal transplant.

## Introduction

Chronic kidney disease (CKD) represents a significant source of disease burden in the United States, with a prevalence of 14% for CKD stages 1–4 and over 20,000 renal transplants performed every year [1, 2]. Cardiovascular disease (CVD) is the leading cause of morbidity and mortality among patients with CKD and renal transplant [3]. Among older individuals the prevalence of CVD is 2-fold higher (69.9% vs 34.7%) in patients with CKD compared with those without CKD [1]. Furthermore, patients with CKD present with a different clinical

**Data Availability Statement:** Data cannot be shared publicly because it was collected for quality purposes and for renal transplant outcome reporting and no patient approval has been

obtained for sharing data. Data that is deintified, are available from the Lifespan Biostatistical Core (contact via Jason Machan jmachan@lifespan.org) for researchers who meet the criteria for access to confidential data.

**Funding:** Dr. Anderson received research funding from Brown University. The funding source played no role in the study design or intepretation.

**Competing interests:** Dr. Abbott is a consultant for Philips and Boston Scientific and has institutional research funding from Abbott Vascular, Sinomed, CSL Behring, Biosensors Research USA. The other authors have nothing to disclose.

profile than the general population; they are often asymptomatic, and conventional cardiac risk factors are less predictive of cardiovascular disease [4]. The American Heart Association/ American College of Cardiology (AHA/ACC) expert consensus guidelines in 2012 recommend routine non-invasive cardiac stress testing for renal transplantation candidates irrespective of symptoms or functional status [5]. However, the cardiology and renal-transplant guidelines are not congruous on this topic and it is unclear if pre-operative stress testing is useful for predicting early and/or late major cardiovascular events (MACE) in patients undergoing renal transplant [6]. We therefore aimed to assess the association between cardiac stress testing results and post-transplant MACE at our institution.

## Methods

In this retrospective single-center cohort study, we selected patients who underwent renal transplant at Rhode Island Hospital, Providence, RI USA from January 2006 to May 2016. Our inclusion criteria were adults age >18 years who underwent renal transplantation and who had cardiac stress testing with imaging within 24 months prior to transplant. Our primary outcome was post-transplant MACE, defined as all-cause mortality, stroke, myocardial infarction, congestive heart failure, and revascularization, at follow-up including in-hospital events. Endpoints were extracted from the electronic health records and institutional renal transplant database.

We extracted data pertaining to patient's demographics, co-morbidities, stress testing, cardiac catheterization as well as clinical endpoints. Eligible stress tests included exercise, pharmacologic, and/or combined exercise-pharmacologic nuclear myocardial perfusion or echocardiographic testing. Stress tests were categorized as normal or abnormal and abnormal results were further graded as mild, moderate, or severe ischemia based on the severity and extent of ischemia using standard criteria [7, 8]. Our patient records provided access to stress reports and not the raw quantitative stress data. In our hospital system, for nuclear stress tests image interpretation was performed incorporating visual and quantitative analysis compared to a gender specific normal database using standardized thresholds for severity and extent of ischemia [9]. For stress echo, wall motion index was scored semi-quantitatively using a 17-segment model [10]. For patients who received multiple stress tests within 24 months prior to transplantation, the most recent test report was used for data extraction.

Categorical variables were expressed as percentages, and continuous variables were expressed as means with standard deviations. Except BMI, none of the variables had missing values >5%. Univariate differences in baseline characteristics between normal and abnormal stress test groups were evaluated using chi2 tests for categorical variables and student's t-test for continuous variables. Univariate and multivariate logistic regression models were used to evaluate the association between abnormal stress test and long-term MACE, but only 30-day MACE rates are reported due to the small number of events. The covariates for multivariate models were selected based on their clinical relevance based on previous studies or those variables with a p<0.1 on the univariate analysis. Associations were examined in a hierarchical model with the following covariates: age, sex, hypertension, diabetes, hyperlipidemia, obesity, current/prior smoking, prior coronary artery disease (CAD), congestive heart failure, atrial fibrillation, and peripheral vascular disease. History of CAD was broadly defined as: known obstructive coronary disease, prior MI, or prior PCI/CABG. Non-obstructive coronary disease by catheterization and coronary CT were not included. All analyses were performed using Stata16.0 (StataCorp, College Station, Texas), and P value < .05 was considered statistically significant. All study procedures were approved by the Lifespan Institutional Review Board at Rhode Island Hospital.

## Results

Overall, the patient population was predominantly white, male, and over 45 years old. Of the 539 patients in our renal transplant database, 240 had undergone stress testing with imaging within 24 months prior to transplant and were eligible for the study. Two hundred and twelve (91.4%) participants underwent nuclear myocardial perfusion, and the remainder underwent stress echocardiography.

Among these, 5 were excluded due to missing MACE data as they were lost to follow-up and 3 for inability to obtain stress test reports from an outside facility, resulting in the final cohort of 232 patients (Fig 1). Of note, 24 patients included in our analysis had inconclusive stress results secondary to submaximal heart rate responses. Given that they had otherwise negative stress results, these patients are included with the normal stress test group. In our cohort, 59 (25.4%) patients had an abnormal stress result, and 173 (74.6%) had a normal stress result. Among those with abnormal result, 45 (76.3%) were graded to have mild, 10 (17.0%) moderate, and 4 (6.8%) severe ischemia.

Table 1 shows baseline demographic and clinical data stratified by patients with normal vs. abnormal stress test. Compared to patients with normal stress test, patients with abnormal

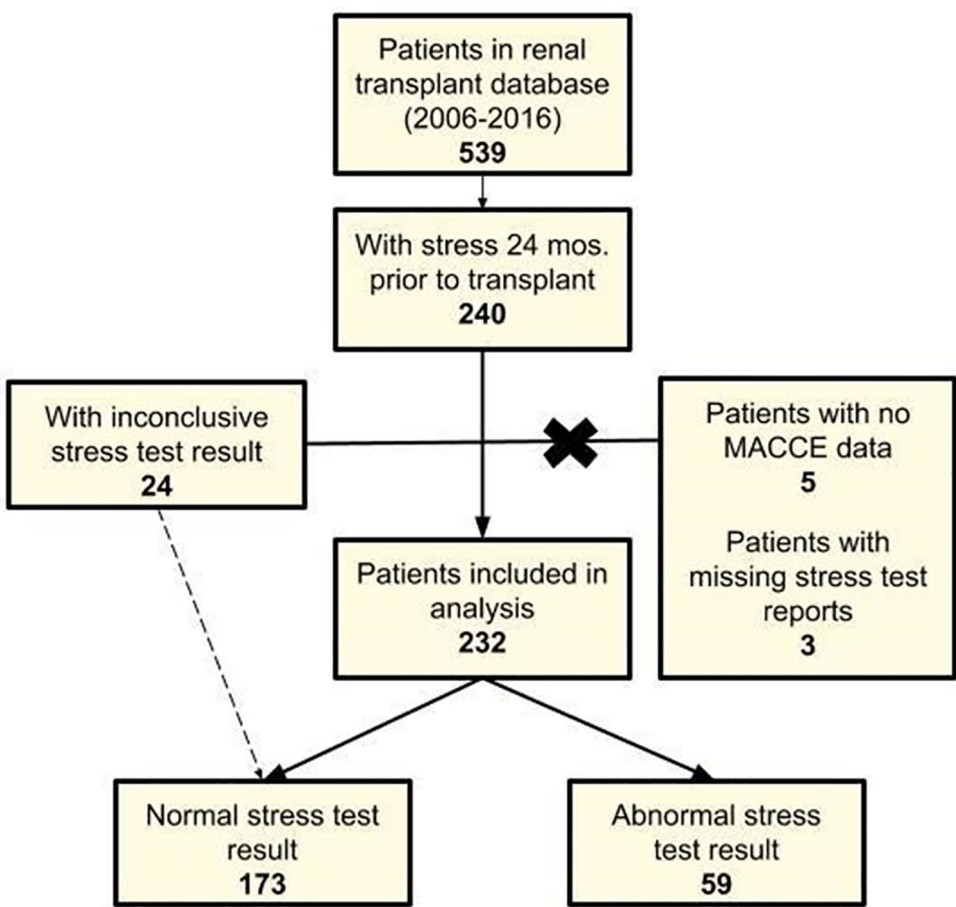

**Fig 1. Flowchart of final patient sample included in analysis.** Flowchart detailing patients included in final data analysis from original identification in database of renal transplant patients. Of the 539 patients in the database, 240 met inclusion criteria (demographic and stress test within 24 months prior to transplant). Of these 240 patients, 8 were excluded because of missing MACE outcomes data or stress test report.

**Table 1. Baseline characteristics of study cohort.**

| | Stress Test | | p value |
|---|---|---|---|
| | **Normal (n = 173)** | **Abnormal (n = 59)** | |
| Age, years | 51.2 ± 12.7 | 53.1 ± 11.9 | 0.31 |
| Women, % | 41.4 | 28.8 | 0.10 |
| Race, % | | | 0.42 |
| White | 68.4 | 77.6 | |
| Black | 19.9 | 8.6 | |
| Hispanic | 7.6 | 8.6 | |
| Asian | 2.9 | 3.5 | |
| Other | 1.2 | 1.7 | |
| Hypertension | 97.1 | 94.8 | 0.41 |
| Dyslipidemia | 57.2 | 72.4 | 0.04 |
| Diabetes mellitus | 38.7 | 58.6 | 0.008 |
| BMI[a], kg/m2 (n = 169) | 27.3 ± 6.3 | 28.9 ± 7.2 | 0.19 |
| Obese | 49.1 | 64.4 | 0.04 |
| Smoking | 42.1 | 28.6 | 0.07 |
| Prior CAD[b] | 15.7 | 52.6 | <0.001 |
| Prior PCI[c] | 2.3 | 27.6 | <0.001 |
| Prior CABG[d] | 3.5 | 10.3 | 0.04 |
| Heart failure | 17.9 | 32.8 | 0.02 |
| Atrial fibrillation | 8.1 | 15.5 | 0.11 |
| PAD[e] | 13.3 | 19 | 0.29 |
| Prior dialysis | 82.6 | 78 | 0.30 |
| Dialysis type | | | 0.99 |
| Hemodialysis | 83.9 | 84.8 | |
| Peritoneal dialysis | 7.0 | 6.5 | |
| Both | 9.1 | 8.7 | |
| Dialysis duration, months (n = 223) | 31.9 ± 40.5 | 26.5 ± 25.2 | 0.36 |
| Prior renal transplant | 17.9 | 13.8 | 0.47 |
| Stress test type | | | |
| Pharmacological | 43.9 | 49.1 | 0.17 |
| Exercise | 52.6 | 42.4 | |
| Exercise + Pharmacological | 3.5 | 8.5 | |

[a]Body mass index.

[b]Coronary artery disease.

[c]Percutaneous coronary intervention.

[d]Coronary artery bypass graft.

[e]Peripheral artery disease.

stress test had higher prevalence of dyslipidemia, diabetes mellitus, obesity, CAD, percutaneous coronary intervention and coronary artery bypass surgery, and heart failure. In our cohort of 232 patients, 9 (3.9%) patients had MACE in the 30-day post-transplant period, 5 of which had an abnormal stress test.

On univariate analysis of long-term outcomes, compared to patients with normal stress test, patients with abnormal stress test had significantly higher rates of MACE (27.8% vs 45.8%, odds ratio (OR) 2.20, 95% CI 1.19–1.04, p = 0.01), but comparable all-cause mortality (7.0% vs 15.3%, p = 0.06). There was no difference in long-term MACE with respect to the

**Table 2. Multivariate analysis for long-term MACE post renal transplant.**

| Variables | MACE (OR, 95% confidence interval) | P value |
|---|---|---|
| Abnormal stress test | 0.83 (0.37–1.92) | 0.68 |
| Age | 1.02 (0.99–1.05) | 0.28 |
| Male | 1.27 (0.63–2.54) | 0.50 |
| Hypertension | 1.22 (0.13–11.6) | 0.86 |
| Diabetes | 2.37 (1.12–5.00) | 0.024 |
| Hyperlipidemia | 0.66 (0.30–1.43) | 0.29 |
| Obese (BMI $\geq$30) | 0.99 (0.52–1.89) | 0.97 |
| Current/prior smoker | 1.10 (0.50–2.05) | 0.98 |
| Coronary artery disease | 3.05 (1.30–7.14) | 0.01 |
| Congestive heart failure | 1.38 (0.61–3.16) | 0.44 |
| Atrial fibrillation | 5.86 (1.86–18.44) | 0.002 |
| Peripheral vascular disease | 1.05 (0.42–2.66) | 0.91 |

MACE: major adverse cardiovascular event, OR: odds ratio, BMI: body mass index.

severity of ischemia. MACE events were 47% (21/45) in patients with mild ischemia, 40% (4/10) with moderate ischemia and 50% (2/4) in severe ischemia group (p = 0.92). On multivariate analysis accounting for clinical risk factors, abnormal stress test was not associated with increased odds of long-term MACE (adjusted OR: 0.83, 95% CI 0.37–1.92, p = 0.68). Of the other clinical risk factors, diabetes (OR 2.37, 95% CI 1.12–5.00, p = 0.02), history of CAD (OR: 3.05, 95% CI 1.30–7.14, p = 0.01) and atrial fibrillation (OR: 5.86, 95% CI 1.86–18.44, p = 0.002) were found to be associated with long-term MACE (Table 2). The results remained consistent when sensitivity analysis was performed combining equivocal/inconclusive stress test results with the abnormal group.

Among patients with an abnormal stress test, 10 were referred for cardiac catheterization (15.6%). The average interval between stress test and catheterization was 3.9 (±3.4) months.

Five had obstructive coronary artery disease of which 2 underwent percutaneous coronary intervention. In addition, 3 patients with normal stress tests were referred for catheterization for typical angina symptoms, and all (100%) had obstructive CAD.

## Discussion

Our retrospective cohort study explored the ability of stress testing to predict MACE in renal transplant patients as well as clinical characteristics independently associated with long-term MACE. The most significant findings from our study were that (1) 30-day MACE rates were low and similar in patients with and without abnormal stress testing and (2) long-term at a median of 5 years, abnormal cardiac stress testing was not independently associated with post-transplant MACE in renal transplant candidates. High risk clinical characteristics that were independently associated with long-term MACE were atrial fibrillation, diabetes mellitus and known CAD.

While used as the primary method for risk stratification, an abnormal stress test did not prevent patients from undergoing renal transplant. This may be related to the fact that stress testing is performed in otherwise suitable transplant candidates and that the majority of abnormal results were graded as mild, with only one quarter classified as moderate or severe. Further, because stress testing was generally performed in our facility as part of a routine pre-operative screening assessment, we have no comprehensive data about patients' symptom burden at time of stress test. It is plausible that many patients with an abnormal stress test who

proceeded to transplant were asymptomatic or had mild symptoms controlled with guideline directed medical therapy. The rate of cardiac catheterization after an abnormal stress test result was very low in our center, since after an abnormal stress test, patients were referred for cardiology consultation, and the risk for transplantation was determined to be acceptable in these largely asymptomatic patients without revascularization. A few patients, however, did undergo revascularization as a result of the work up and went on to transplantation.

In our study, the overall MACE rate was high at median follow-up of 5 years and abnormal stress test was not associated with MACE. The prognostic value of stress test for predicting hard clinical events varies based on patient's age, clinical and symptom profile. The inability of an abnormal stress test result to predict MACE suggest that conditions other than obstructive CAD, such as atrial fibrillation or microvascular disease, which are not as reliably detected by stress testing, or new plaque rupture, which is inherently unpredictable, may have played a substantial role in prognosis in patients with advanced CKD and renal transplantation.

The 2012 AHA Consensus Statement on Cardiac Disease Evaluation and Management Among Kidney and Liver Transplant Candidates states that "Non-invasive stress testing may be considered in kidney transplant candidates with no active conditions on the basis of the presence of multiple CAD risk factors regardless of functional status (Class IIb, Level of Evidence C)" [5]. The strength of this recommendation has been weakened by its low evidence rating, lack of consensus among professional groups in other specialties, and lack of confirmatory studies since the recommendation was published. Our study supports the growing body of literature that questions the routine use of stress testing as a risk stratifying tool in this patient population. In a propensity-matched study, Dunn et al. [11] examined the utility of stress testing in patients with no active ischemic disease within 18 months of renal transplant. Their results showed no association between performance of stress testing and all-cause mortality, total MI, and fatal MI at 30 days post-transplant, suggesting that stress testing does not have a role in predicting peri-operative events [9]. Our study expands on this data by categorizing the results of the stress testing and extending the follow-up period to several years and supports their results in confirming a lack of association at long-term follow-up. Our 30-day event rates were similar to those reported by Dunn et al. [11]; however, a smaller number of overall events precluded multivariate analysis for 30-day follow-up. As an alternative to stress testing, the study by Park et al. suggests that among relatively young patients with good functional capacity and shorter dialysis duration, transthoracic echocardiography may be as effective as stress testing in predicting ischemic heart disease pre-operatively in this population [12]. Others argue that the poor sensitivity, specificity, and positive predictive value of non-invasive methods make them significantly inferior to coronary angiography, and question whether stress testing should be used at all for low-to-moderate risk patients [13]. The CARP trial [14] has notably demonstrated that cardiac revascularization prior to elective vascular surgery does not affect the rate of long-term mortality or MI, suggesting that invasive methods of risk stratification offer increased risk but limited utility. Majority of patients in our study did not routinely undergo cardiac catheterization after an abnormal stress test and were managed with optimal medical therapy. Recently, in a post-hoc analysis from the ISCHEMIA-CKD trial [15] of 194 patients with chronic coronary syndromes and at least moderate ischemia on stress testing who are listed for renal transplant, an invasive strategy did not improve outcomes compared with conservative medical management.

There is some literature that supports the use of stress testing, especially among certain higher risk patient populations. Doukky et al. [16] assessed the prognostic utility of the 8 risk factors set forth by the AHA/ACCF consensus statement and the role of noninvasive stress testing in this context. The authors reported that patients with 3–4 risk factors derive the greatest additional prognostic benefit from myocardial perfusion imaging, but stress testing per se

was not predictive of post-operative MACE. The Doukky et al. [16] study specifically included 8 risk factors outlined by the AHA/ACCF; our study includes all of these risk factors except left ventricular hypertrophy. The long-term follow-up period was similar (median 4.7 vs 5 years), but our event rate is higher (32.3% vs. 27.1%); however, our MACE included the additional variables of congestive heart failure and revascularization.

Our study has several limitations. This is a retrospective study where data was extracted from review of medical records spanning multiple years and hence subjected to missing or incomplete data as well as variability in documentation. Additionally, we cannot account for patient encounters that occurred outside of our health care facility. Patients' symptom status was unknown at the time of stress testing, as any testing within the eligible timeframe was assumed to be part of pre-transplant evaluation. Our study population was predominantly limited to white males, warranting further investigation into the risk patterns of other racial/ethnic groups. Finally, eligible patients in our single-center study were derived from a database of patients who successfully received renal transplant, indicating an overall healthier patient population than those with CKD/ESRD who did not undergo transplant. We did not have data available to accurately estimate the number of patients who were removed from consideration of transplantation due to abnormal stress testing results. However, we estimate the number is low due to current practice of cardiology consultation with medical optimization and referral for cardiac catheterization in selected patients that may benefit from percutaneous or surgical revascularization prior to transplantation. Our results therefore may not be generalizable to patients with more significant disease burden.

In conclusion, our results indicate that an abnormal stress test is not predictive of long-term MACE post-renal transplant. Additionally, the high event rate in patients with negative stress tests suggests that we should not be reassured by a normal test result. Our study does not support routine stress testing, particularly in presumably asymptomatic individuals, who are undergoing renal transplantation.

## Author Contributions

**Conceptualization:** Kelsey Anderson, J. Dawn Abbott.

**Data curation:** Kelsey Anderson, Dhaval Kolte, Reginald Gohh, James A. Arrighi, Philip Stockwell, J. Dawn Abbott.

**Formal analysis:** Chirag Bavishi, Dhaval Kolte.

**Investigation:** Kelsey Anderson, Philip Stockwell.

**Methodology:** Chirag Bavishi, Dhaval Kolte, Reginald Gohh, J. Dawn Abbott.

**Supervision:** J. Dawn Abbott.

**Writing – original draft:** Kelsey Anderson.

**Writing – review & editing:** Chirag Bavishi, Dhaval Kolte, Reginald Gohh, James A. Arrighi, Philip Stockwell, J. Dawn Abbott.

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
