## [Decision Letter · Decision Letter 0]

13 Aug 2021

PONE-D-21-21764

Relation of abnormal cardiac stress testing with outcomes in patients undergoing renal transplantation

PLOS ONE

Dear Dr. Abbott,

Thank you for submitting your manuscript to PLOS ONE. After careful consideration, we feel that it has merit but does not fully meet PLOS ONE’s publication criteria as it currently stands. Therefore, we invite you to submit a revised version of the manuscript that addresses the points raised during the review process. Reviewers felt that the long term outcome data, which is a major asset of the study, could be further developed. The Editor agrees with this sentiment.  Inclusion of a survival curves and performance of multivariable analysis of long-term outcome would significantly elevate the impact of the manuscript. 

We look forward to receiving your revised manuscript.

Kind regards,

Jeffrey J. Rade, MD

Academic Editor

PLOS ONE

Journal Requirements:

2. Thank you for providing the date(s) when patient medical information was initially recorded. Please also include the date(s) on which your research team accessed the databases/records to obtain the retrospective data used in your study.

"Dr. Anderson received research funding from Brown University."

Reviewers' comments:

Reviewer's Responses to Questions

**Comments to the Author**

1. Is the manuscript technically sound, and do the data support the conclusions?

Reviewer #1: Yes

Reviewer #2: Partly

2. Has the statistical analysis been performed appropriately and rigorously? 

Reviewer #1: Yes

Reviewer #2: Yes

3. Have the authors made all data underlying the findings in their manuscript fully available?

Reviewer #1: Yes

Reviewer #2: Yes

4. Is the manuscript presented in an intelligible fashion and written in standard English?

Reviewer #1: Yes

Reviewer #2: Yes

5. Review Comments to the Author

Reviewer #1: This is a very interesting an pertinent paper regarding the role of routine stress testing prior to renal transplantation.

The authors have a large cohort of patients that had renal transplantation. They found that the stress test result did not correlate with 30 or 5 year outcomes (MACE). DM, CAD, and AF did correlate with increased risk of adverse events.

This is an important study that adds to a growing body of literature showing that routine stress testing prior to renal transplantation is not helpful. Please see my specific comments below.

1. Study was predominantly limited to white males. This should be mentioned in the limitation section of the discussion.

2. It is unclear to me if the 30 day MACE included events from the index hospitalization during which the transplant was performed. This should be clarified in the methods section.

3. Page 6. Were the patients who had cardiac catheterization included in the analysis?

Reviewer #2: The authors undertook this single center study of patients undergoing renal transplantation. This is a common group of patients to receive routine stress tests, and so this is an important investigation to determine if the stress test results were independently associated with MACE.

The authors found, in multivariable adjusted models, that diabetes, coronary artery disease, and atrial fibrillation were independently associated with 30-day MACE, but abnormal stress test results were not.

Suggestions/considerations for revision:

1) How many people were evaluated for renal transplant at your center and not transplanted due to stress testing results? Or a way to estimate this number?

2) I am wondering how you graded abnormal exercise treadmill tests as mild, moderate, or severe ischemia? You discuss using exercise, pharm, or a combined approach, but don’t discuss the modality (echo, nuclear, treadmill only).

3) I can’t tell if you are looking at MACE at 30 days or a different time frame based on your methods/results. You say 30-day MACE in the abstract but not the body of the paper.

4) Table 2 does not include the covariates adjusted for – I assume it was all the covariates listed in your methods paragraph 3?

5) How did you define prior history of coronary artery disease? Could this include non-obstructive disease seen on cath or coronary CT? I see that you included PCI/prior CABG as a co-variate, but I don’t see that listed in Table 2.

6) You mention sensitivity analysis performed combining equivocal/inconclusive stress tests. Have you looked at other cut-points with the abnormal stress tests, such as those with moderate or severe ischemia only to determine associations with MACE?

7) What about outcomes beyond 30 days? It looks like you have 5 years of follow up data. Did you follow these participants longer to look at associations beyond 30 days?

8) When you present your results, I assume you are presenting odds ratio (2.20) on the last line of page 5, but it is not labelled.

9) Participants who undergo pharmacologic testing because of inability to exercise/use a treadmill are associated with higher adverse outcomes that those undergoing exercise testing. Do you specifically stratify those undergoing pharm vs. exercise testing?

10) I am not sure your conclusion “particularly in asymptomatic individuals” is supported by your study, since you don’t know what symptoms these patients had. I know that is inferred based on the data you present, but probably best to keep that out of your last line. Or qualify to say “presumably asymptomatic”

Minor typos to correct – you go between MACE and MACCE throughout the paper, with MACCE seen in the abstract, on page 6, and Figure 1. First line of page 6 – “and but comparable all-cause mortality” Second line page 6 “with respect of severity of ischemia”. Page 10 – “abnormal stress test, ,”.

6. PLOS authors have the option to publish the peer review history of their article (what does this mean?). If published, this will include your full peer review and any attached files.

Reviewer #1: **Yes: **Timothy P. Fitzgibbons

Reviewer #2: No

---

## [Author Response · Author response to Decision Letter 0]

14 Oct 2021

Editors’ comments:

Reviewers felt that the long-term outcome data, which is a major asset of the study, could be further developed. The Editor agrees with this sentiment. Inclusion of a survival curves and performance of multivariable analysis of long-term outcome would significantly elevate the impact of the manuscript

We are glad that the editors find the long-term outcomes of our study to be of importance. The 30 day and long term (median follow up 5 years) MACE rates and multivariable analysis for the long-term outcomes are now presented. No adjusted outcomes for 30-day events are presented due to the small number of events. We were unable to include survival curves as we do not have the exact date of events. The renal transplant dataset at our institution tracks events over time periods, which include a range of time. 

Reviewer 1 comments: 

1. Study was predominantly limited to white males. This should be mentioned in the limitation section of the discussion.

Thank you for your comment. We have updated the limitation section to include the following: “Our study population was predominantly limited to white males, warranting further investigation into the risk patterns and utility of stress testing in other racial/ethnic groups.”

2. It is unclear to me if the 30-day MACE included events from the index hospitalization during which the transplant was performed. This should be clarified in the methods section.

Yes, all events from the time of transplant, including in-hospital were recorded in the renal transplant dataset. This detail was added to the methods. 

3. Page 6. Were the patients who had cardiac catheterization included in the analysis?

Yes, in our cohort 10 patients with abnormal stress testing underwent cardiac catheterization and 3 had revascularization with PCI prior to transplantation. Patients that had abnormal stress testing that did not go on to transplantation, however, were not captured in this analysis. This was acknowledged in response to one of the comments from reviewer 2 and the manuscript updated accordingly. 

Reviewer 2 comments:

1) How many people were evaluated for renal transplant at your center and not transplanted due to stress testing results? Or a way to estimate this number? 

Our patient population was selected from a database that only included patients who successfully underwent transplant, and the data dates to 2006. Unfortunately, we do not have data available to accurately estimate those patients who were not captured in our target population due to stress testing results. In review with our renal transplant team, they estimate very few patients were excluded from transplantation based on stress testing results. The rationale is that stress testing is performed in otherwise suitable candidates and if abnormal cardiology consultation obtained and the patient is medically managed or referred for cardiac catheterization. If PCI or CABG is required, patients are generally relisted for transplantation 6 months after revascularization. A situation where stress testing uncovers CAD is advanced and not amenable to revascularization is rare at our center. “We did not have data available to accurately estimate the number of patients who were removed from consideration of transplantation due to abnormal stress testing results. However, we estimate the number is low due to current practice of cardiology consultation with medical optimization and referral for cardiac catheterization in selected patients that may benefit from percutaneous or surgical revascularization prior to transplantation.”

2) I am wondering how you graded abnormal exercise treadmill tests as mild, moderate, or severe ischemia? You discuss using exercise, pharm, or a combined approach, but don’t discuss the modality (echo, nuclear, treadmill only).

Thank you for pointing out that the stress testing modality was not presented. The stress tests were nuclear myocardial perfusion imaging studies (91.4%) and echo (8.6%). In addition to indicating the stress testing modality we added the following details regarding grading abnormality to the methods. “Our patient records provided access to stress reports and not the raw quantitative stress data. In our hospital system, for nuclear stress tests image interpretation was performed incorporating visual and quantitative analysis compared to a gender specific normal database, using standardized thresholds for severity and extent of ischemia. For stress echo wall motion index was scored semi-quantitatively using 17 segment model.” References for stress test interpretation and reporting were added. 

We have updated the results section to include the above statement. “Two hundred and twelve (91.4%) participants underwent nuclear perfusion scan; the remainder underwent stress echo.” 

3) I can’t tell if you are looking at MACE at 30 days or a different time frame based on your methods/results. You say 30-day MACE in the abstract but not the body of the paper. 

We apologize that the primary outcome data presentation was not clear and that we erroneously reported adjusted 30 day rather than long tern MACE rates in the abstract. In our study we report MACE at 30 days post-transplant, as well as long-term follow up at a median (IQR) follow-up duration of 5 (3-7) years. There were only 9 events at 30-day MACE, therefore we report only the event rates of these outcomes between normal and abnormal stress test result. MACE at long term follow-up were analyzed with multivariate analysis. 

We have updated the abstract to reflect the correct data and added to the methods section “Univariate and multivariate logistic regression models were used to evaluate the association between abnormal stress test and long-term MACE, but only 30-day MACE rates are reported due to the small number of events.” Our results section includes the following, which has been revised to include the number of patients at 30-day MACE with abnormal stress test: “In our cohort of 232 patients, 9 (3.9%) patients had MACE in the 30-day post-transplant period, 5 of whom had an abnormal stress test. Seventy-five (32.3%) patients had MACE at a median (IQR) follow up duration of 5 (3-7) years.” 

4) Table 2 does not include the covariates adjusted for – I assume it was all the covariates listed in your methods paragraph 3?

Thank you for pointing out this omission. Variables with a p<0.1 on the univariate analysis (Table 1) or deemed clinically significant (age, AF, PVD) were included. To avoid collinearity, prior CAD but not prior PCI/CABG was included in the model. This detail was added to the methods. 

5) How did you define prior history of coronary artery disease? Could this include non-obstructive disease seen on cath or coronary CT? I see that you included PCI/prior CABG as a co-variate, but I don’t see that listed in Table 2.

Prior PCI/CABG was included in the univariate analysis, but not used in regression analysis due to inclusion of CAD in the model; consequently ‘Prior PCI/CABG’ is intentionally absent from Table 2. History of CAD was broadly defined as: known obstructive coronary disease, prior MI, or prior PCI/CABG. Non-obstructive coronary disease seen on catheterization and coronary CT findings were not included. 

We have removed “prior history of revascularization (PCI or CABG)” from the list of covariates in the methods section. 

Additionally, we have updated the Methods section by defining CAD.

6) You mention sensitivity analysis performed combining equivocal/inconclusive stress tests. Have you looked at other cut-points with the abnormal stress tests, such as those with moderate or severe ischemia only to determine associations with MACE?

Due to the small sample size of patients with moderate to severe ischemia we were underpowered to analyze this subgroup separately. The severity of ischemia, however, was not significantly associated with unadjusted rates of long-term MACE. In the results “There was no difference in long term MACE with respect to the severity of ischemia. MACE events were 47% (21/45) in patients with mild ischemia, 40% (4/10) with moderate ischemia and 50% (2/4) in severe ischemia group (p=0.92)”. Also, in the limitations we acknowledge that results may not apply to patients with greater burden of disease than our population. 

7) What about outcomes beyond 30 days? It looks like you have 5 years of follow up data. Did you follow these participants longer to look at associations beyond 30 days?

Sorry if this was unclear (see response to comment 3). In our study we examine MACE at 30 days post-transplant, as well as long-term follow up at a median (IQR) follow-up duration of 5 (3-7) years. The abstract, methods and results were reviewed to assure this was clear. We removed reference to MACCE as the endpoints were MACE at 30 days and long term. 

8) When you present your results, I assume you are presenting odds ratio (2.20) on the last line of page 5, but it is not labelled.

Thank you for pointing out this omission, we have added that 2.2 refers to the odds ratio (OR) on univariate logistic regression. 

9) Participants who undergo pharmacologic testing because of inability to exercise/use a treadmill are associated with higher adverse outcomes that those undergoing exercise testing. Do you specifically stratify those undergoing pharm vs. exercise testing?

While we acknowledge that inability to perform exercise is a potential predictor or poor outcome, we did not perform analysis stratified by type of stress testing to avoid overanalyzing the data and inability to control for confounding in choice of stress modality. 

10) I am not sure your conclusion “particularly in asymptomatic individuals” is supported by your study, since you don’t know what symptoms, these patients had. I know that is inferred based on the data you present, but probably best to keep that out of your last line. Or qualify to say “presumably asymptomatic”

We have updated the manuscript to the following: “Our study doesn’t support routine stress testing, particularly in presumably asymptomatic individuals, who are undergoing renal transplantation.” 

Minor typos to correct – you go between MACE and MACCE throughout the paper, with MACCE seen in the abstract, on page 6, and Figure 1. First line of page 6 – “and but comparable all-cause mortality” Second line page 6 “with respect of severity of ischemia”. Page 10 – “abnormal stress test,”.

We thank the reviewer for taking the time to provide such detailed comments. We recognize the inconsistences and grammatical errors and have corrected them.

---

## [Decision Letter · Decision Letter 1]

16 Nov 2021

Relation of abnormal cardiac stress testing with outcomes in patients undergoing renal transplantation

PONE-D-21-21764R1

Dear Dr. Abbott,

We’re pleased to inform you that your manuscript has been judged scientifically suitable for publication and will be formally accepted for publication once it meets all outstanding technical requirements.

Kind regards,

Jeffrey J. Rade, MD

Academic Editor

PLOS ONE

Additional Editor Comments (optional):

Reviewers' comments:

Reviewer's Responses to Questions

**Comments to the Author**

1. If the authors have adequately addressed your comments raised in a previous round of review and you feel that this manuscript is now acceptable for publication, you may indicate that here to bypass the “Comments to the Author” section, enter your conflict of interest statement in the “Confidential to Editor” section, and submit your "Accept" recommendation.

Reviewer #1: All comments have been addressed

Reviewer #2: All comments have been addressed

2. Is the manuscript technically sound, and do the data support the conclusions?

Reviewer #1: Yes

Reviewer #2: Yes

3. Has the statistical analysis been performed appropriately and rigorously? 

Reviewer #1: Yes

Reviewer #2: Yes

4. Have the authors made all data underlying the findings in their manuscript fully available?

Reviewer #1: Yes

Reviewer #2: Yes

5. Is the manuscript presented in an intelligible fashion and written in standard English?

Reviewer #1: Yes

Reviewer #2: Yes

6. Review Comments to the Author

Reviewer #1: All my concerns have been addressed. Paper is very well written, succinct and clear. Discussion is excellent.

Reviewer #2: Thank you for your thoughtful revisions. My last comment would be interpreting your results in the context of the Ischemia study.

7. PLOS authors have the option to publish the peer review history of their article (what does this mean?). If published, this will include your full peer review and any attached files.

Reviewer #1: **Yes: **Timothy P. Fitzgibbons MD PhD

Reviewer #2: No

---

## [Editor Report · Acceptance letter]

22 Nov 2021

PONE-D-21-21764R1 

Relation of abnormal cardiac stress testing with outcomes in patients undergoing renal transplantation 

Dear Dr. Abbott:

I'm pleased to inform you that your manuscript has been deemed suitable for publication in PLOS ONE. Congratulations! Your manuscript is now with our production department. 

Kind regards, 

on behalf of

Dr. Jeffrey J. Rade 

Academic Editor

PLOS ONE